

**Comparing the spatial patterns of climate change in the 9[th] and 5[th] millennia B.P. from**
**TRACE-21 model simulations**
**Liang Ning[1,2,3], Jian Liu[1,2*], Raymond S. Bradley[3], and Mi Yan[1,2]**
[1]Key Laboratory of Virtual Geographic Environment, Ministry of Education; State key Laboratory
of Geographical Environment Evolution, Jiangsu Provincial Cultivation Base; School of
Geographical Science, Nanjing Normal University, Nanjing, 210023, China
[2]Jiangsu Center for Collaborative Innovation in Geographical Information Resource Development
and Application, Nanjing, 210023, China
[3]Climate System Research Center, Department of Geosciences, University of Massachusetts,
Amherst, 01003, United States
[*]jliu@njnu.edu.cn
**ABSTRACT**
The spatial patterns of global temperature and precipitation changes, as well as corresponding
large-scale circulation patterns during the latter part of the 9[th] and 5[th] millennia B.P. (4800-4500
versus 4500-4000 years B.P. and 9200-8800 versus 8800-8000 years B.P.) are compared through
a group of transient simulations using Community Climate System Model version 3 (CCSM3).
Both periods are characterized by significant sea surface temperature decreases over the North
Atlantic south of Iceland.  Temperatures were also colder across the northern hemisphere, but
warmer in the southern hemisphere. Significant precipitation decreases are seen over most of the
northern hemisphere, especially over Eurasia and the Asian monsoon regions, indicating a weaker
summer monsoon. Large precipitation anomalies over northern South America and adjacent ocean
regions are related to a southward displacement of the Inter Tropical Convergence Zone (ITCZ).
Climate changes in the late 9[th] millennium B.P. ("The 8.2ka BP event ") are widely considered to
have been caused by a large fresh water discharge into the northern Atlantic, which is confirmed
in a meltwater forcing sensitivity experiment, but this was not the cause of changes occurring
between the early and latter half of the 5[th] millennium B.P. We speculate that long-term changes
in insolation related to precessional forcing led to cooling, which passed a threshold around 4500
years B.P., leading to a reduction in the Atlantic meridional overturning circulation (AMOC) and
associated teleconnections across the globe. The onset of the Neoglacial occurred around this time,



and the subsequent changes in glacierization have persisted, modulated by internal centennial-
scale ocean-atmosphere variability. We suggest that the "4.2ka B.P. event" was one of several late
Holocene multi-century fluctuations that were embedded in a longer-term, lower frequency change
in climate, linked to orbital forcing.

**1.  Introduction**
It is well-documented that the first order driver of Holocene climate change was orbital
forcing, with an overall decline in summer insolation in summer months, particularly at high
latitudes.  This led to a drop in temperatures at high latitudes and less rainfall throughout the
monsoon regions of the northern hemisphere, as seen in many paleoclimatic records (Burns, 2011;
Solomina et al., 2015). Shorter-term rainfall fluctuations around this long-term change in
hydrological conditions are clearly seen in many speleothem and lacustrine sediment records (e.g.
Wang et al., 2005; Kathayat et al., 2017).  Abrupt hydrological changes around 4.2 ka BP have
been documented for various regions of the world (Weiss, 2016). For example, based on a variety
of physical and biological proxies, Booth et al. (2005) found a severe drought that affected the
mid-continent of North America between 4.1 and 4.3 ka BP.  Over eastern Asia, Tan et al. (2018)
found that there were droughts over the northern part of eastern China while floods occurred over
the southern part around 4.2 ka BP. It was suggested that these climate changes may have been
global in extent (Bond et al., 2001; Thompson et al., 2002; Booth et al., 2005).
In recent years, a more comprehensive picture of the 4.2 ka BP event has been derived
from analysis of new high-resolution proxy data from different regions, and the event has become
the focus of several symposia and research conferences (Weiss, 2016).  This event is of particular
interest as it is associated with societal collapse and regional abandonment in many different
regions.  For example, the collapse and abandonment of Akkadian imperial settlements in the
Khabur Plains, and other communities in dry farming domains across the Aegean and West Asia,
was in response to the abrupt nature with which the megadrought began (with its onset in less than
five years), its magnitude (a precipitation reduction of 30-50%) and its long duration (200-300
years) (Weiss, 2015).
Although a drought episode around 4.2ka B.P. has been found in many proxy
reconstructions, the mechanisms that brought this about are still unclear though different
hypotheses have been proposed.  For example, Staubwasser and Weiss (2006) suggested that the





abrupt climate change event at 4.2ka B.P., as well as other widespread droughts around 8.2ka BP
and 5.2ka BP over the eastern Mediterranean, West Asia, and the Indian subcontinent, were caused
by a change in subtropical upper-level flow over the eastern Mediterranean and Asia. Weiss (2016)
also suggested that the major global monsoon and ocean-atmosphere circulation systems were
deflected or weakened synchronously at 4.2ka BP, causing major century-scale precipitation
disruptions (severe megadroughts) over different regions. Other studies (Wang et al., 2005; Tan
et al., 2018) have also noted weakening of the Asian summer monsoon at around this time,
resulting in drought over the northern part of eastern China and flooding over the southern part.

Some studies have suggested that these large-scale circulation anomalies may be induced
by persistent modes of internal climate variability, though there is a wide range of explanations.
For example, Booth et al. (2005) indicated that the widespread mid-latitude and subtropical
drought around 4.2ka BP was linked to a La Niña-like SST pattern, possibly associated with
amplification of this spatial mode by variations in solar irradiance or volcanism. On the other
hand, Hong et al. (2005) analyzed a 12000-yr proxy record for the East Asian monsoon and
concluded that such abnormal climate conditions could possibly result from frequent and severe
El Niño activities. Using paired oxygen isotope records from North America, Liu et al. (2014b)
indicated that there was a transition from a negative Pacific North American (PNA)-like pattern
during the mid-Holocene to a positive PNA-like pattern during the late Holocene, which led to
drier conditions in northwestern North America. A similar conclusion was reached by
Finkenbinder et al. (2016) based on lake sediment records from Newfoundland. They argued that
this transition took place around 4.3ka B.P., leading to wetter conditions across the Newfoundland
region. In contrast, Bond et al. (2001) argued that North Atlantic SST anomalies around 4.2ka
B.P. were related to the North Atlantic Oscillation (NAO) pattern. Deininger et al. (2017) also
found that changes in the atmospheric circulation associated with northward and southward
propagating westerlies (similar to the NAO but on a millennial instead of decadal scale) could be
a possible driver of coherency and cyclicity during the last 4.5ka BP, as seen in multiple
speleothem $\delta^{18}$O records that span most of the European continent. However, the ultimate driver
for this oscillation remains unclear.

Wang (2009a) reviewed studies of Holocene cold events, and concluded that an outburst
flood in which pro-glacial Lake Agassiz drained a large volume of freshwater into the North
Atlantic extremely rapidly (Bianchi and McCave, 1999; Risebrobakken et al., 2003; McManus et



al., 2004; Clarke et al., 2004) initiated the cold 8.2ka BP event, leading to a brief reorganization
of the North Atlantic Meridional Overturning Circulation (AMOC).  Potential external forcing
factors for the 4.2ka BP event include non-linear responses to Milankovitch forcing, solar
irradiance variations, and explosive volcanic eruptions, all of which may have brought about
variations in the ocean-atmosphere system (Booth et al., 2005).  Wang (2009a) concluded that
solar irradiance minima were the main cause of cold events in the mid- to late Holocene (including
the 4.2ka BP event) and that oscillations within the climate system could possibly have intensified
these cold events under certain circumstances (Wang, 2009b).  Bond et al. (2001) also proposed a
possible link between the 4.2ka BP event and reduced solar radiation at that time.

In summary, the 8.2ka BP event and corresponding shifts in the ITCZ were caused by

glacial flooding of the North Atlantic and this can be reasonably simulated by coupled GCMs with
different boundary conditions and freshwater forcings (Alley and Agustsdottir, 2005; LeGrande et
al., 2006). At 4.2ka B.P., the major global monsoon and ocean-atmosphere circulation systems
may have been deflected or weakened synchronously, causing major century-scale precipitation
disruptions, resulting in severe megadroughts over many different regions (Weiss, 2016).
However, the forcing mechanisms that brought about the 4.2ka BP event are currently uncertain.
GCM simulations of the 4.2ka BP event have not received much attention, therefore, in this study,
the spatial patterns and corresponding mechanisms relevant to the 4.2 ka BP event and 8.2ka BP
event are compared.

**2.    Data and methodology**
The simulations of the last 21ka (TRACE-21) were used in this study (He, 2011; He et al. 2013).
These transient simulations have been completed using Version 3 of the Community Climate
System Model (CCSM3), which is a coupled atmosphere-ocean general circulation model
developed by National Center for Atmospheric Research (NCAR).  The atmosphere model in the
CCSM3 is the Community Atmospheric Model 3 (CAM3) with a horizontal resolution ~3.75°
(T31), and the ocean model is the Parallel Ocean Program (POP) with a longitudinal resolution of
3.6° and variable latitudinal resolution.  The "full-forcing" TRACE-21 simulation includes
changes in orbital parameters, greenhouse gases, ice extent (based on the ICE 5G-VM2
configurations) and meltwater fluxes from the Northern Hemisphere and Antarctic ice sheets.
Simulations in which only one of these factors was included have also been carried out and are





available in the TRACE-21 archive (Otto-Bliesner et al., 2006). These simulations can reproduce
the timing and magnitude of many aspects of climate evolution during the last 21 ka, such as sea
surface temperature (SST) (He et al., 2013).  However, there are significant differences between
the rate of temperature change in the model during the early Holocene and many paleoclimatic
records (Liu et al., 2014a; Marcott et al., 2013).  In this study, we do not address this enigma, but
use the transient model data to compare intervals within the Holocene when abrupt changes in
climate are known to have occurred in some regions (~8.2ka B.P. and ~4.2ka B.P.).  These times
were recently adopted by the International Commission on Stratigraphy as the chronological
boundaries of the early, mid and late Holocene (Walker et al., 2012). We examine mean annual
surface temperature, annual precipitation and SSTs from the full-forcing experiment.
**2.1 Results**

First, we assess Holocene climate variability as simulated in the full-forcing experiment.  Fig.

1 shows the time series of surface temperature and precipitation over the last past 13ka.  It shows
cooling associated with the Younger Dryas, followed by Holocene warming, but also a brief
cooling episode from ~8500-8000 B.P.  Thereafter the record exhibits strong multi-century scale
variability. Temperature and precipitation are positively correlated at this global scale.  It is
tempting to associate the colder episodes with those identified by Wanner et al (2011) or by Bond
et al. (2001) but only a few of these are coincident in time.

The period 4.5ka-4.0ka BP was chosen for analysis, by subtracting the mean annual 2m air

temperatures, SSTs and precipitation from 4500-4000 years B.P. from the preceding period (4800-
4500 years B.P.). The spatial distribution of air temperature (Fig. 2a) shows that temperatures were
significantly colder over most of the extra-tropical northern hemisphere, but generally warmer in
the Tropics and in the southern hemisphere.  The main exception is in northern South America,
which was cooler, and northern India and Pakistan, which were significantly warmer. Precipitation
decreased over almost all of the northern hemisphere, particularly in the Tropics where the ITCZ
shifted south, resulting in higher rainfall in the 0-20ºS zonal band from 4500-4000 years B.P.
(Figure 2b). There was less precipitation over the northern part of China but more precipitation
over southern China, consistent with paleoclimate reconstructions that indicate a weaker East
Asian monsoon (Wang et al., 2005). This pattern is similar to some of the megadroughts that have
happened in recent centuries (Cook et al., 2010).  Over other Asian monsoon regions, such as
India, there were also significant precipitation reductions during the second half of the 5[th]



millennium B.P., consistent with speleothem records that show a decline in Indian summer
monsoon rainfall over this period (Kathayat et al., 2017). Over Central America and the northern
edge of South America, conditions were also drier in the later period, but over the rest of South
America, and adjacent ocean regions, precipitation was higher due to a southward displacement of
the ITCZ; this pattern is supported by speleothem records of rainfall in Mexico and Brazil
(Lachniet et al., 2013; Bernal et al., 2016).   The SST pattern shows significantly cooler
temperatures in the period 4500-4000years B.P. over the North Atlantic. This cooling is centered
around 50°N (south of Iceland) and extends into the sub-Tropics on the eastern side of the sub-
tropical gyre.   Slightly cooler temperatures are also found over the North Pacific (Fig. 2c).   By
contrast, for most of the southern hemisphere there was a positive change in temperature.  Rotated
EOF analysis on the global SST field shows the primary feature (in EOFs 1 and 2) to be the cooler
SSTs over the North Atlantic, with a shift around 4.5ka BP from a predominantly positive to a
generally negative pattern (Fig. 3).  This is similar to an AMO-like pattern over the northern
Atlantic that has been identified in both instrumental and paleoclimatic records (Delworth and
Mann, 2000; Knudsen et al., 2011).

The same evaluation of changes in the 9[th] millennium B.P. was made by subtracting the

mean annual 2m air temperatures, SSTs and precipitation from 8800-8000 from the preceding
period (9200-8800 years B.P.).  Air temperatures were significantly lower in the second period
over most of the northern hemisphere; only a zone from northern South America across to sub-
Saharan Africa and India was warmer in the second period (Fig. 4a). Almost the entire southern
hemisphere was warmer.  Precipitation was less in the second period across all of the northern
hemisphere, especially along the ITCZ, which was displaced to the south.  This resulted in an
increase in rainfall in a belt south of the Equator, across almost all of the Tropics (Fig. 4b).  The
rest of the southern hemisphere was also slightly wetter.  SSTs show a strong pattern of cooling
over the North Pacific, and the eastern North Atlantic, south of Iceland, extending around the
Atlantic sub-tropical gyre into the tropical Atlantic and Caribbean (Fig. 4c). Rotated EOFs show
that the anomalies in the North Atlantic and North Pacific dominate the first 3 EOFs (Fig. 5).

The spatial patterns of temperature changes, precipitation changes, and SST changes were

remarkably similar in the late 9[th] millennium as in the period leading up to the late 5[th] millennium
(Fig. 6).   The major difference (Fig. 6a) is that SST changes over the subtropical Atlantic were
greater and the related changes across the northern hemisphere in the 9[th] millennium B.P. were



larger than in the 5$^{th}$ millennium. Similarly, the major changes in precipitation patterns were
comparable, but less pronounced from 4500-4000 years B.P. These similarities are somewhat
puzzling as the meltwater forcing sensitivity experiment clearly shows that the "8.2ka BP event"
was induced by a massive freshwater flux into the Atlantic whereas (as far as we know) no
comparable meltwater event occurred in the late Holocene so it seems unlikely that such forcing
was a factor driving the changes seen in the model output for 4500-4000 years B.P. A possible
explanation is that as summer insolation at high latitudes of the northern hemisphere declined over
the Holocene, a threshold was passed which led to cooler SSTs in the North Atlantic and a
consequent reduction in the Atlantic meridional overturning circulation (AMOC), with
teleconnections into the southern hemisphere. In our experiment, we examined just the 5th
millennium B.P., but it is possible that the changes seen in the latter half of the period were more
persistent, and typical of the rest of the Holocene (the Neoglacial). Indeed, there is much evidence
for cooler conditions and glacier expansion around the North Atlantic around this time (Solomina
et al., 2015). Thereafter, glaciers fluctuated but did not disappear again, indicating that a different
climate state prevailed. Fluctuations around these cooler mean conditions may be related to
internal centennial-scale ocean-atmosphere variability (cf. Wanner et al., 2011). This is distinctly
different from the period prior to 5000 years B.P. when many mountain regions were ice-free.
Further analysis of the TRACE21 simulations are needed to fully explore this matter.

**3.  Conclusions**
Paleoclimate records have shown that cold and dry conditions persisting for several centuries
around 4.2ka BP over many different regions and these had devastating societal impacts. In this
study, the spatial patterns of temperature, precipitation, and corresponding circulation anomalies
during the latter part of the 9$^{th}$ and 5$^{th}$ millennia B.P. (4800-4500 versus 4500-4000 years B.P. and
9200-8800 versus 8800-8000 years B.P.) were compared based on model simulations. The
changes in climate during both periods were similar and characterized by significant temperature
and precipitation decreases over most of the northern hemisphere, but the southern hemisphere
was slightly warmer and wetter. In particular, the ITCZ was displaced to the south and monsoon
regions of the northern hemisphere were generally drier. On a regional scale, there was less
precipitation over the northern part of China but more precipitation over southern China, indicating
a reduced eastern Asian summer monsoon in the period 4500-4000 years B.P.



It is clear that the earlier period was strongly influenced by freshwater forcing in the North
Atlantic, but this can not explain the changes in the 5$^{th}$ millennium B.P.  We speculate that long-
term changes in insolation related to precessional forcing led to cooling, which passed a threshold
around 4500 years B.P., leading to a reduction in the AMOC and associated teleconnections across
the globe. Based on widespread paleoclimatic evidence for the onset of neoglaciation (Solomina
et al., 2015), it seems clear that there was a fundamental shift in climate around this time.
Furthermore, those changes have persisted, with minor fluctuations, through to the present.
Interestingly, SSTs in the area of the North Atlantic where cooling was so prominent from 4500-
4000 years B.P. do show multi-century-scale oscillations for the remainder late Holocene, with
temperatures below the 4800-4500 years B.P. average for ~69% of the time (Fig. 7). Whether such
changes are also linked to hydrological anomalies elsewhere, as with the period 4500-4000 years
B.P., is not known, but it seems likely, given the large-scale coherent link between temperature
and precipitation that is apparent in Fig. 1. Whether such fluctuations reflect internal centennial-
scale ocean-atmosphere variability, or external forcing (explosive volcanic eruptions and/or solar
irradiance forcing) is also not known. Further studies of the role of external forcing are needed to
provide a better understanding of such mechanisms (cf. Ottera et al., 2010; Gupta and Marshall,
2018).  Nevertheless, we conclude from the model simulations that the "4.2ka B.P. event" was
simply one of several late Holocene multi-century fluctuations that were embedded in a longer-
term, lower frequency change in climate resulting from orbital forcing.

**Acknowledgements**
This research was jointly supported by the National Key Research and Development Program of
China (Grant No. 2016YFA0600401), the National Natural Science Foundation of China (Grant
No. 41501210, Grant No. 41420104002, Grant No. 41671197, and Grant No. 41631175), the
Jiangsu Province Natural Science Foundation (Grant No. BK20150977), Top-notch Academic
Programs Project of Jiangsu Higher Education Institutions (Grant No. PPZY2015B115), and the
Priority Academic Development Program of Jiangsu Higher Education Institutions (Grant No.
164320H116).  It also received support from U.S. NSF grant PLR-1417667 to the University of
Massachusetts. TraCE-21ka was made possible by the DOE INCITE computing program, and
supported by NCAR, the NSF P2C2 program, and the DOE Abrupt Change and EaSM programs.

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





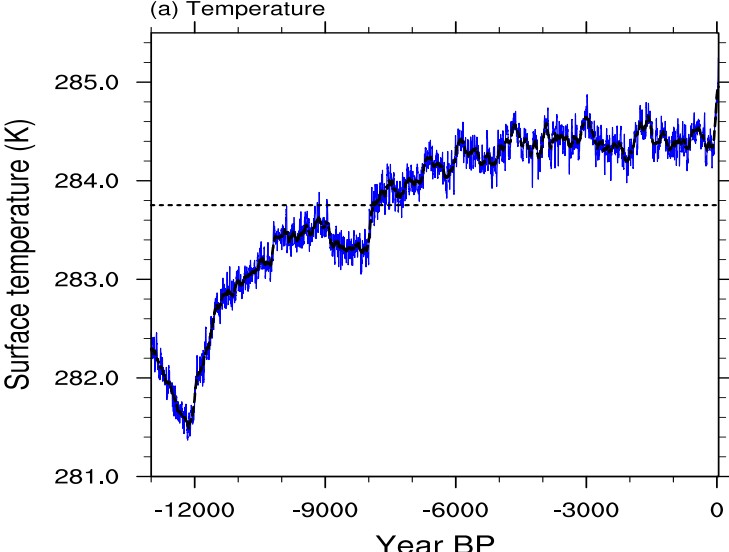


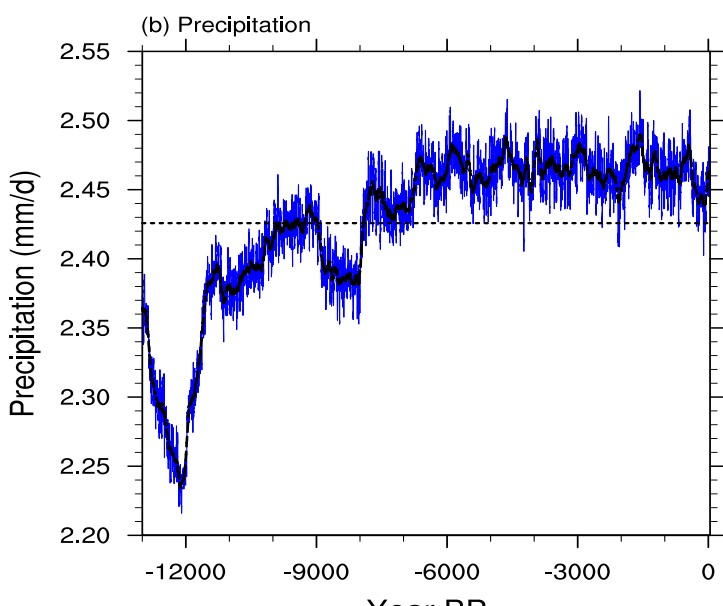

**Figure 1**. The 10-year running averaged (blue line) and 100-year running averaged (black line)
northern hemisphere average surface temperature and precipitation over the last 13ka years from
the all-forcing experiment.



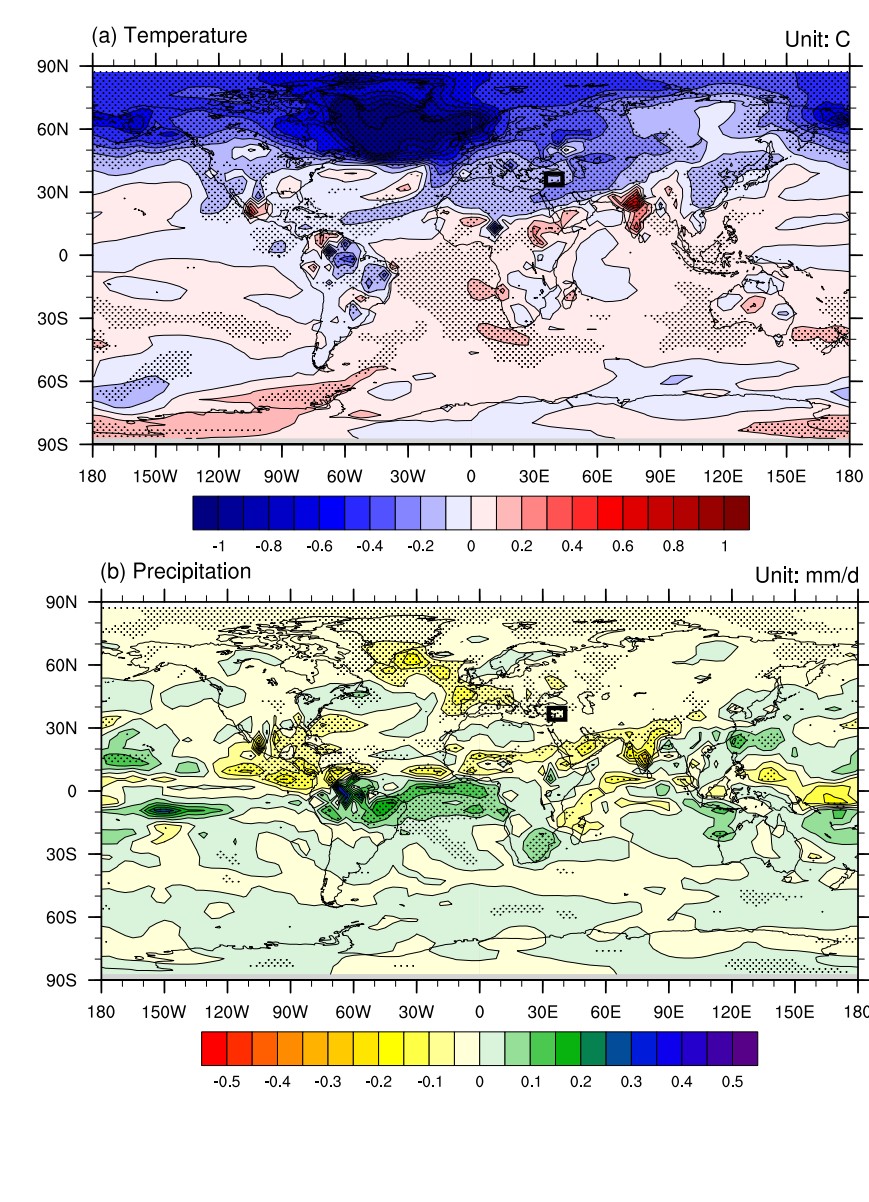






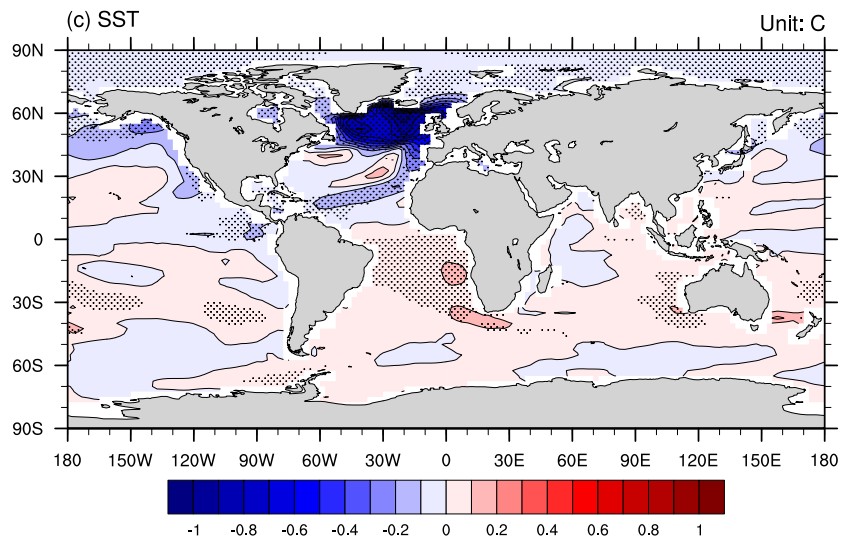

**Figure 2**. The changes of surface temperature (a, unit: °C), precipitation (b, unit: mm/day), and
SST (c, unit: °C) after 4.5ka BP (between 4500-4000ka BP and 4800-4500ka BP)
The rectangles indicate the region with major dry-farming settlement abandonments shown in
Fig. 2 of Weiss (2016)


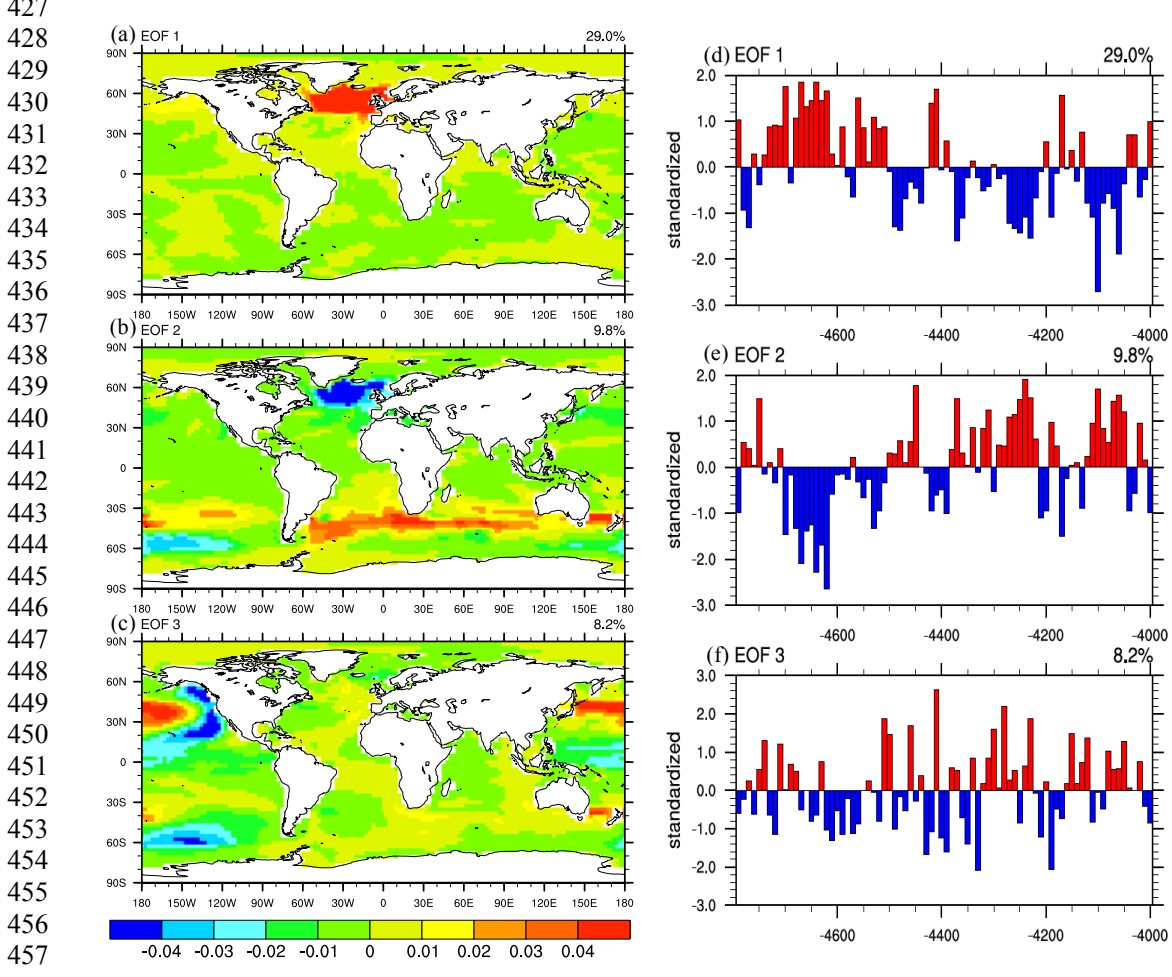

**Figure 3**. The first three patterns (a-c) and principal components (d-f) of rotated EOF modes on
the SST over the period 4800-4000ka BP.






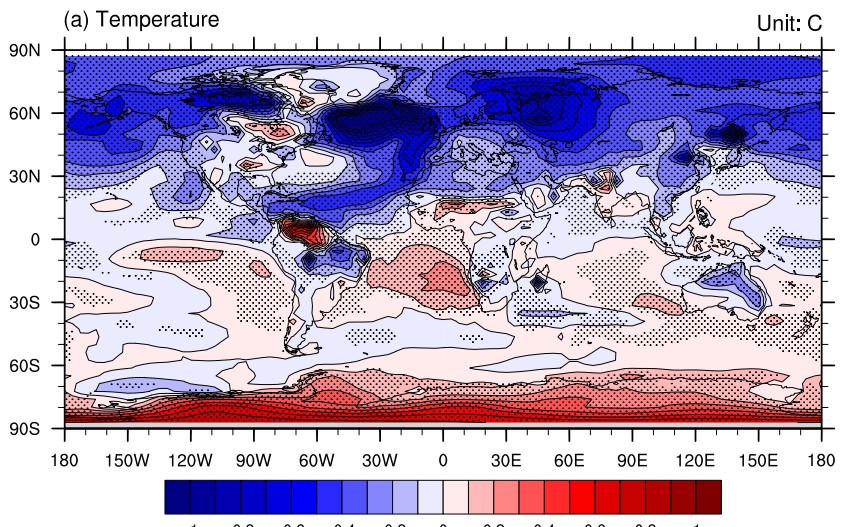


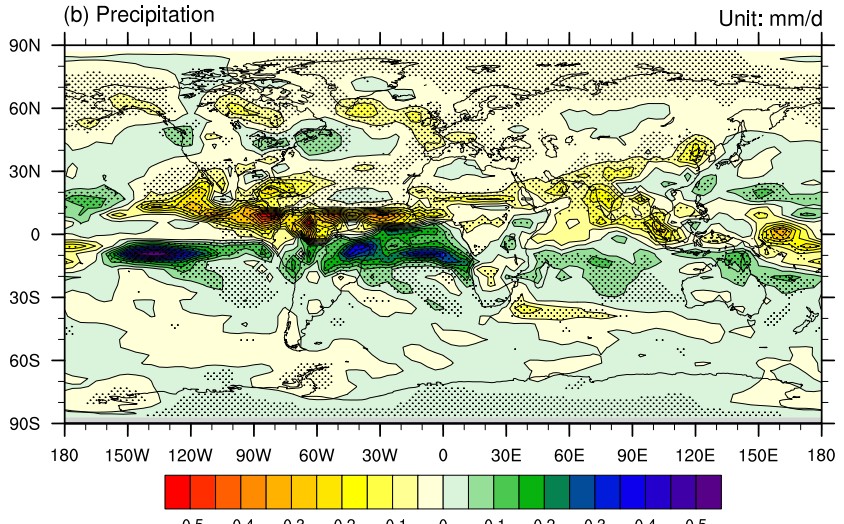






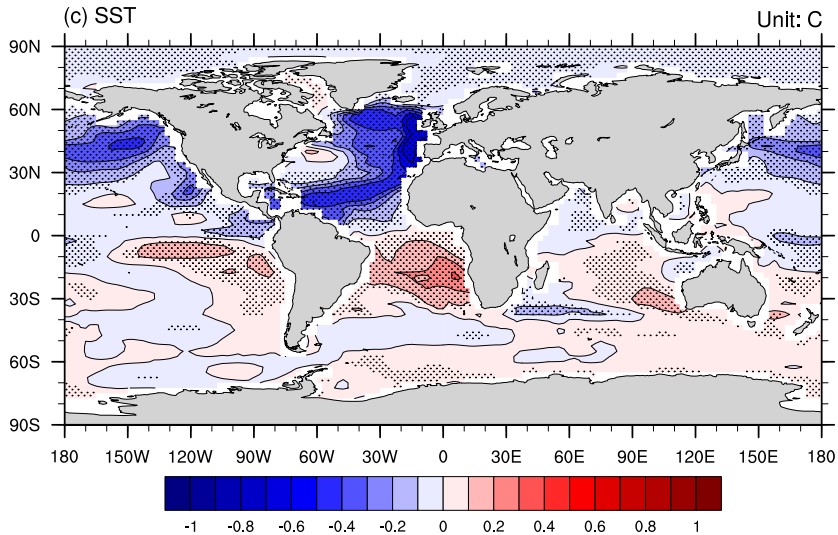

**Figure 4**. The changes of surface temperature (a), (a, unit: °C), precipitation (b, unit: mm/day),
and SST (c, unit: °C) after 8.8ka BP (between 8800-8000ka BP and 9200-8800ka BP)





**Figure 5**. The first three patterns (a-c) and principal components (d-f) of rotated EOF modes on
the SST over the period 9200-8000ka BP.




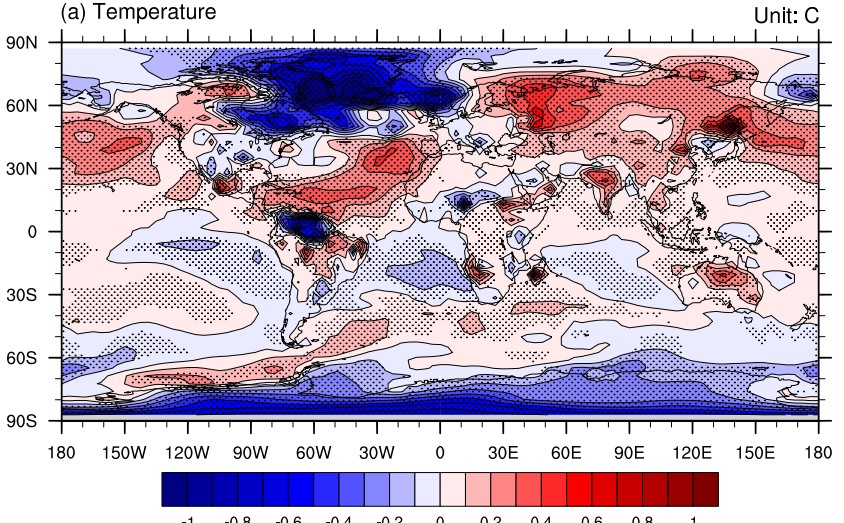


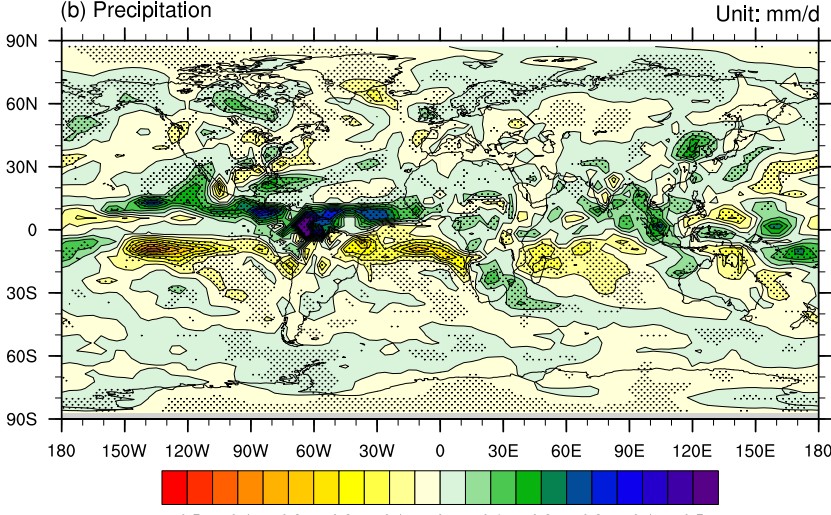






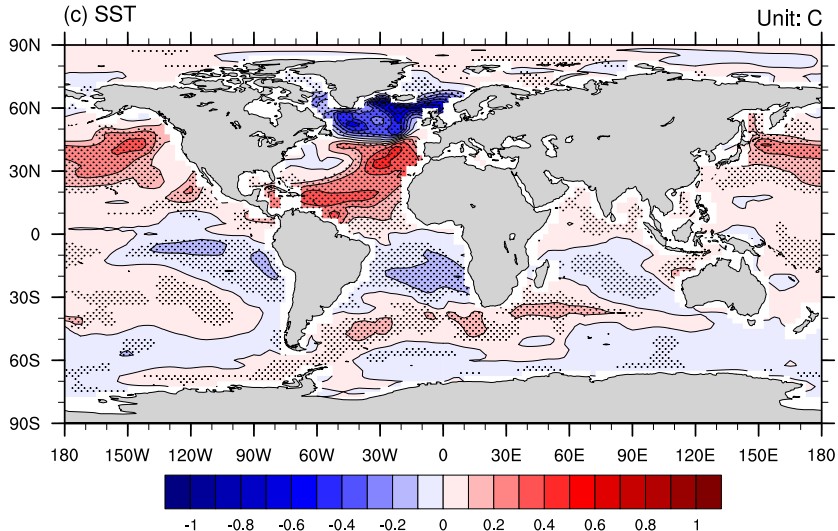

**Figure 6**. The differences between changes of surface temperature (a), (a, unit: °C), precipitation (b, unit: mm/day), and SST (c, unit: °C) of the 5$^{th}$ millennium BP and 9$^{th}$ millennium BP periods shown in Figures 2 and 4.

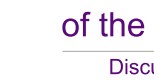
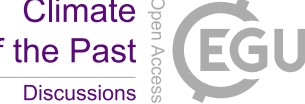

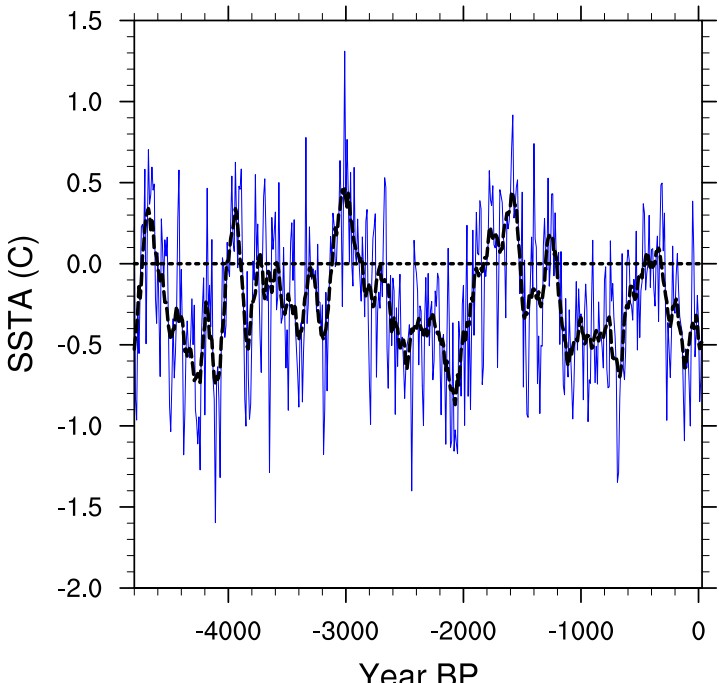

**Figure 7.**  SSTs in the area of the North Atlantic shown in dark blue (40-60 °N, 7.5-60 °W) on
Figure 2c, plotted as anomalies from the mean for 4800-4500 years B.P.  ~69% of the time,
temperatures in this region were below the mean.