# Peer review of "Comparing the spatial patterns of climate change in the 9[th] and 5[th] millennia B.P. from TRACE-21 model simulations"

_Climate of the Past, 2018_

## Referee Comment (RC1) · Anonymous Referee #1 · 4 Nov 2018

In this paper, Ning et al. studied the spatial patterns of temperature, precipitation, and circulation anomalies during the latter part of the 9th and 5th millennium B.P. by using model simulations. They suggested that the long-term decline of insolation caused the cooling of North Atlantic passing a threshold around 4500 years B.P., and lead to a reduction in the AMOC and associated teleconnections across the globe. The result will help us to a better understanding of the 4.2 ka event. I think this is a very good paper and could be published in CP after minor revisions. Here are my comments and suggestions. 1. line16-17: I can't understand this kind of discription. You are discussing the climate change during the late 9th and 5th millennia BP, but use 9200-8800 versus 8800-8000a BP, 4800-4500 versus 4500-4000 a BP to defined them. It makes me con-

fused. 2. The English is generally good, however, I think it could still be benifit from a native English speaker. For example, line 42, "around" better be "superimpose"; line 61: "about" should be "drought"; line 62: "have" should be "had"... 3. line 65-70: here talk about the record of 4.2 ka drought. I suggest to move this paraghraph to the end of the first paragraph. 4. line 85: positive NAO, or negative NAO? 5. line 143-145: unclear. Do you mean the temperature during (4800-4500 a BP) minus temperature during (4500-4000 a BP) ? or the inverse? 6. line 152-154: consistent with paleoclimate reconstructions (Tan et al., 2018, EPSL) that indicate a weaker East Asian monsoon (Wang et al., 2005). This pattern is similar to the situation during the LIA in China (Tan et al., 2018, QSR), and some of the megadroughts happened in recent centuries (Cook et al., 2010). 7. line 172-173, why do you choose 8.8 ka as a dividing line? why not 8.5 ka? 8. line 188, revise "from" to "during"?

---

## Editor Comment (EC1) · H. Weiss (Editor) · 4 Nov 2018

Ning et al paper overlaps with Mi Yan et al paper, using same model simulation (TraCE-21ka) to refine our knowledge of the 4.2 ka BP event, its spatio-temporal dynamics, and causes. The paper, however, actually clouds our knowledge of the 4.2 ka BP event to that extent that it probably should be withdrawn for publication.

The paleoclimate data that inform this model simulation are not presented, listed or discussed. Simulation results are only generally described, and have little relevance to the distribution of paleoclimate proxy data for the 4.2 ka BP event (probably more than 300 synchonous records). The simulation takes 4.5 ka BP as a start date for the event,

which is erroneous, and then finds that there is no event.

The event started at ca. 4.3-4.2 ka BP in globally distributed, regionally coherent, high resolution terrestrial records. That is, synchronous records for 2-300 years drought, with only slight interruptions in some records, i.e., 3-stages, occur at 4.3 -4.2 ka BP, with a few minor and constrained regions experiencing unusual abrupt wet or cold events. These more than 300 proxy records, or another 200 lower resolution records, do not present this event beginning at 4.5 ka BP, nor at 3.7 ka BP.

Specifically, the precise data for megadrought across the NA Great Plains, across the Mediterranean from Spain to Turkey, across the Iranian plateau, across the Indian Summer Monsoon domains both in India and NE Africa, and across Tibet and north China, across Africa, and down the SA Andes, comprise more than 300 synchronous high resolution proxy records. Some isolated regions, such as the north Atlantic and NE Pacific had synchronous abrupt wet or cold events. We already know this happened at ca 4.2/4.3 ka BP, not at 4.5 ka BP.

---

## Referee Comment (RC2) · Anonymous Referee #2 · 5 Nov 2018

In this manuscript, the authors compared the spatial patterns of global temperature, precipitation, and SST during two centennial-scale droughts during the Holocene based on model simulation. The similarities and differences between these two drought events, which are believed to be caused by different reasons, are examined in details. The authors also hypothesized that the drought during the 5th millennium B.P. is caused by a reduction in the AMOC due to the long-term changes in insolation related to precessional forcing, which passed a threshold around 4.5 ka B.P. This manuscript covers two important topics: one topic is the detailed spatial patterns during the 4.2 ka BP event, which could be used for comparison with proxy reconstructions, and the other topic is mechanisms behind the 4.2 ka B.P., which are interesting to the whole

paleoclimate community. So, I believe this manuscript should be interesting to a wide audience of Climate of the Past. Some interesting results and meaningful conclusions are shown in this manuscript, and the analyses are straightforward and clear, however, I still have some comments regarding the manuscript listed below. Therefore, I would recommend that the present manuscript may be accepted for publication after some minor revisions. 1. The numbering of the manuscript needs to be re-arranged, for example "Results" should be Section 3 rather than Section 2.1. 2. More details of the TRACE-21 experiments should be provided for the readers, such as the external forcing used in the experiments. 3. The authors claim that the 4.2 ka BP event was one of the several late Holocene centennial-scale fluctuations, have they compared the timing of these fluctuations with the Bond events? Do they have some similiarities? 4. Line 198, considering the 5th millennium BP event as the start of the Neoglacial is a really interesting topic, which should be strengthened with more discussion. 5. In Fig. 1, the dash lines are the means, right? The authors should add this information into the caption. 6. In the figure captions, the time "4500 ka BP" should be "4.5 ka BP", and also other similar timings. 7. In the caption of Fig. 7, the phase "shown in dark blue" is obscure, and should be revised.

---

## Short Comment (SC1) · 7 Nov 2018

Yan et al examine TRACE-21 GCM simulations and conclude that internal variability explains climatic anomalies at the beginning of the 5th Millennium B.P. Ning et al also examine TRACE-21 simulations and find that there is a pattern of air temperature, precipitation & SST anomalies similar to those seen in the 9th Millennium B.P., which resulted from freshwater forcing and a shutdown of the AMOC. Freshwater forcing in the 5th Millennium is not a plausible explanation for the similarity in the climatic anomaly patterns and so some other mechanism is required. Ning et al conclude that precessional forcing is the most likely explanation, noting that the area of maximum cooling in

the North Atlantic in the early 5th Millenium BP remains cold through most of the later Holocene, with only minor fluctuations. It is possible that such minor fluctuations result from internal variability superimposed on the long-term orbitally-driven changes, and so there is nothing inherently incompatible between the conclusions of Ning et al and Yan et al. Given that 2 reviewers suggest only minor revisions to the Ning et al manuscript, the suggestion that the paper be withdrawn is unacceptable. Reviewer #1 notes that "this is a very good paper and could be published in CP after minor revisions"; reviewer #2 states that: "I would recommend that the present manuscript may be accepted for publication after some minor revisions". We are willing to revise the paper as reviewers suggest, and can cross-reference to Yan et al's paper, but do not accept that the paper is unsuitable for Climate of the Past. The fact that we compared a longer interval (4000-4500) with the preceding interval has no bearing on our conclusions; it is quite clear from Figure 7 that the anomaly around 4.2ka B.P. is part of a longer period of cooling, and is similar to other anomalies that occurred over the last few millennia. Our goal was not to prove or reinforce conventional dogma about a purported "4.2ka B.P. event" but to objectively examine a long-term model simulation that spans this period of time. The model simulation does not rely on specific paleoclimate records, so it should not be surprising that such data are "not presented, listed or discussed". It is clear from the analysis we present that there were areas with anomalously low precipitation, and other areas that were unusually wet in the late 5th millennium B.P. It would involve a much more extensive analysis to compare specific sites with model-derived estimates of P-E, and that is surely something that could be carried out in the future. However, the claim that there are 300 synchronous high-resolution proxy records indicating drought at 4.2ka B.P. (and another 200 at lower resolution. . .?) is preposterous fiction.

---

## Author Response (AR1)

**Reviewer #1**

In this paper, Ning et al. studied the spatial patterns of temperature, precipitation, and circulation anomalies during the latter part of the 9th and 5th millennia B.P. by using model simulations. They suggested that the long-term decline of insolation caused the cooling of North Atlantic passing a threshold around 4500 years B.P., and lead to a reduction in the AMOC and associated teleconnections across the globe. The result will help us to a better understanding of the 4.2 ka event. I think this is a very good paper and could be published in CP after minor revisions. Here are my comments and suggestions.

We really appreciate the valuable comments and suggestions from the reviewer. In this revision, we carefully addressed all the concerns from the reviewer, and we hope that the reviewer finds this revision satisfactory.

1. line16-17: I can't understand this kind of discription. You are discussing the climate change during the late 9th and 5th millennia BP, but use 9200-8800 versus 8800-8000a BP, 4800-4500 versus 4500-4000 a BP to defined them. It makes me confused.

In this study, one major motivation is to compare the spatial patterns from cold event due to external forcing ("The 8.2ka BP event") with cold event due to internal variability superimposing on long-term decline ("The 4.2ka BP event"). Because the model cannot reproduce the exact timing of the cold events as the reconstruction, we can only select the timing with temperature decrease around the 8.2ka BP and 4.2ka BP in the simulation to represent these two events.

2. The English is generally good, however, I think it could still be benifit from a native English speaker. For example, line 42, "around" better be "superimpose"; line 61: "about" should be "drought"; line 62: "have" should be "had"...

Thank you for these suggestions, but the text as written is correct.

3. line 65-70: here talk about the record of 4.2 ka drought. I suggest to move this paraghraph to the end of the first paraghraph.

We amended the first paragraph to improve the discussion.

4. line 85: positive NAO, or negative NAO?

Negative NAO. We added this information in line 85.

5. line 143-145: unclear. Do you mean the temperature during (4800-4500 a BP) minus temperature during (4500-4000 a BP) ? or the inverse?

The differences between the two periods mean the temperature during period (4500-4000 a BP) minus the period (4800-4500 a BP). We clarify this in the manuscript.

6. line 152-154: consistent with paleoclimate reconstructions (Tan et al., 2018, EPSL) that indicate a weaker East Asian monsoon (Wang et al., 2005). This pattern is similar to the situation during the LIA in China (Tan et al., 2018, QSR), and some of the megadroughts happened in recent centuries (Cook et al., 2010).

We appreciate the reviewer providing this information.
We have added the discussion into the manuscript, and also cited the corresponding references.

7. line 172-173, why do you choose 8.8 ka as a dividing line? why not 8.5 ka?
From the temperature time series (Fig. 1a) the abrupt changes occurred around 8.8 ka BP, and this
timing is also confirmed by the first principal component of REOF analysis on the SST (Fig. 4a).
Therefore, we chose 8.8 ka BP as the dividing line. We added this clarification into the manuscript.
8. line 188, revise "from" to "during"?
No change made.

**Reviewer #2**

In this manuscript, the authors compared the spatial patterns of global temperature, precipitation, and SST during two centennial-scale droughts during the Holocene based on model simulation. The similarities and differences between these two drought events, which are believed to be caused by different reasons, are examined in details. The authors also hypothesized that the drought during the 5th millennium B.P. is caused by a reduction in the AMOC due to the long-term changes in insolation related to precessional forcing, which passed a threshold around 4.5 ka B.P.

This manuscript covers two important topics: one topic is the detailed spatial patterns during the 4.2 ka BP event, which could be used for comparison with proxy reconstructions, and the other topic is mechanisms behind the 4.2 ka B.P., which are interesting to the whole paleoclimate community. So, I believe this manuscript should be interesting to a wide audience of Climate of the Past. Some interesting results and meaningful conclusions are shown in this manuscript, and the analyses are straightforward and clear, however, I still have some comments regarding the manuscript listed below. Therefore, I would recommend that the present manuscript may be accepted for publication after some minor revisions.

We really appreciate the valuable comments and suggestions from the reviewer. We have carefully addressed all these concerns, and we hope that the reviewer finds this revision satisfactory.

1. The numbering of the manuscript needs to be re-arranged, for example "Results" should be Section 3 rather than Section 2.1.

The numbering of the manuscript has been re-arranged. The "Results" is now Section 3, and the "Discussion and Conclusions" is now Section 4.

2. More details of the TRACE-21 experiments should be provided for the readers, such as the external forcing used in the experiments.

Following the reviewer's suggestion, we added the following information "*The orbital forcing is based on transient variations of orbital configuration (Berger, 1978). The concentrations of greenhouse gases were adopted from study of Joos and Spahni (2008). The ice sheet data were modified from the reconstruction of Peltier (2004). The meltwater scheme was adopted from study of Liu et al. (2009)*" in to the second paragraph of Section 2.

3. The authors claim that the 4.2 ka BP event was one of the several late Holocene centennial-scale fluctuations, have they compared the timing of these fluctuations with the Bond events? Do they have some similiarities?

Both of them have similar centennial-scale variability but the timing of "Bond events" does not match the fluctuations seen in the model simulations.

4. Line 198, considering the 5th millennium BP event as the start of the Neoglacial is a really interesting topic, which should be strengthened with more discussion.

Following the reviewer's suggestion, we now added more discussion into the manuscript.

We also added a new Fig. 7 to show that the AMOC has been decreasing since 4.5 ka BP, especially in the orbital forcing only simulation (new Fig. 7b).

5. In Fig. 1, the dash lines are the means, right? The authors should add this information into the caption.

The reviewer is correct & we have added "*The black dash lines show the averages of the time*

*series*" to the caption.
6. In the figure captions, the time "4500 ka BP" should be "4.5 ka BP", and also other similar
timings.
The figure captions have been changed to "Year" to be consistent with the x-axis ranges.
7. In the caption of Fig. 7, the phase "shown in dark blue" is obscure, and should be revised.
The caption has been revised as "*the area of the North Atlantic with significant negative SST*
*differences between the the 5$^{th}$ millennium BP and 9$^{th}$ millennium BP periods (40-60 °N, 7.5-60*
*°W)*" to be clearer.

**Comparing the spatial patterns of climate change in the 9[th] and 5[th] millennia B.P. from TRACE-21 model simulations**

**Liang Ning[1,2,3], Jian Liu[1,2*], Raymond S. Bradley[3], and Mi Yan[1,2]**

[1]Key Laboratory of Virtual Geographic Environment, Ministry of Education; State key Laboratory of Geographical Environment Evolution, Jiangsu Provincial Cultivation Base; School of Geographical Science, Nanjing Normal University, Nanjing, 210023, China

[2]Jiangsu Center for Collaborative Innovation in Geographical Information Resource Development and Application, Nanjing, 210023, China

[3]Climate System Research Center, Department of Geosciences, University of Massachusetts, Amherst, 01003, United States

[*]jliu@njnu.edu.cn

**ABSTRACT**

The spatial patterns of global temperature and precipitation changes, as well as corresponding large-scale circulation patterns during the latter part of the 9[th] and 5[th] millennia B.P. (4800-4500 versus 4500-4000 years B.P. and 9200-8800 versus 8800-8000 years B.P.) are compared through a group of transient simulations using the Community Climate System Model version 3 (CCSM3). Both periods are characterized by significant sea surface temperature decreases over the North Atlantic south of Iceland. Temperatures were also colder across the northern hemisphere, but warmer in the southern hemisphere. Significant precipitation decreases are seen over most of the northern hemisphere, especially over Eurasia and the Asian monsoon regions, indicating a weaker summer monsoon. Large precipitation anomalies over northern South America and adjacent ocean regions are related to a southward displacement of the Inter Tropical Convergence Zone (ITCZ) in that region. Climate changes in the late 9[th] millennium B.P. ("The 8.2ka BP event ") are widely considered to have been caused by a large fresh water discharge into the northern Atlantic, which is confirmed in a meltwater forcing sensitivity experiment, but this was not the cause of changes occurring between the early and latter half of the 5[th] millennium B.P. Model simulations suggest that a combination of factors, led by long-term changes in insolation, drove a steady decline in SSTs across the North Atlantic and a reduction in the AMOC, over the past 4500 years, with associated teleconnections across the globe, leading to drought in some areas. Multi-century scale fluctuations in SSTs and AMOC strength were superimposed on this decline. This helps explain the onset of neoglaciation around 5000-4500 BP, followed by a series of neoglacial advances and retreats during recent millennia. The "4.2ka B.P. event" appears to have been one of several late Holocene multi-century fluctuations that were embedded in the long-term, low frequency change in climate that occurred after ~4.8 ka BP. Whether these multi-century fluctuations were a response to internal centennial-scale ocean-atmosphere variability or external forcing (such as explosive volcanic eruptions and associated feedbacks) or a combination of such conditions, is not known and requires further study.

**1.    Introduction**

It is well-documented that the first order driver of Holocene climate change was orbital forcing, with an overall decline in summer insolation in summer months, particularly at high latitudes. This led to a drop in temperatures at high latitudes and less rainfall throughout the monsoon regions of the northern hemisphere, as seen in many paleoclimatic records (Burns, 2011; Solomina et al., 2015). Shorter-term rainfall fluctuations superimposed on this long-term change in hydrological conditions are clearly seen in many speleothem and lacustrine sediment records (e.g. Wang et al., 2005; Kathayat et al., 2017). Abrupt hydrological changes around 4.2 ka BP have been documented for various regions of the world; it has been suggested that the major global monsoon and ocean-atmosphere circulation systems were deflected or weakened synchronously at this time, causing major century-scale precipitation disruptions (severe megadroughts) over different regions (Weiss, 2017). Other studies (Wang et al., 2005; Tan et al., 2018a) have also noted weakening of the Asian summer monsoon at around this time, resulting in drought over the northern part of eastern China and flooding over the southern part.

In recent years, a more comprehensive picture of the "4.2 ka BP event" has been derived from analysis of new high-resolution proxy data from different regions, and the event has become the focus of symposia and research conferences (e.g. Weiss, 2015). This event is of particular interest as it is associated with societal collapse and regional abandonment in many different regions. For example, the collapse and abandonment of Akkadian imperial settlements in the Khabur Plains, and other communities in dry farming domains across the Aegean and West Asia, was in response to the abrupt nature with which the megadrought began (with its onset in less than five years), its magnitude (a precipitation reduction of 30-50%) and its long duration (200-300

years) (Weiss, 2017).

Although a drought episode around 4.2ka B.P. has been found in many proxy reconstructions, the mechanisms that brought this about are still unclear, though different hypotheses have been proposed. For example, Staubwasser and Weiss (2006) suggested that the abrupt climate change event at 4.2ka B.P., as well as other widespread droughts around 8.2ka BP

and 5.2ka BP over the eastern Mediterranean, West Asia, and the Indian subcontinent, were caused by a change in subtropical upper-level flow over the eastern Mediterranean and Asia. Some studies have suggested that these large-scale circulation anomalies may reflect persistent modes of internal climate variability, though there is a wide range of other explanations. For example, Booth et al.

(2005) indicated that the widespread mid-latitude and subtropical drought around 4.2ka BP was linked to a La Niña-like SST pattern, possibly associated with amplification of this spatial mode by variations in solar irradiance or volcanism. On the other hand, Hong et al. (2005) analyzed a

12,000-yr proxy record for the East Asian monsoon and concluded that such abnormal climate conditions could possibly result from frequent and severe El Niño activities. Using paired oxygen isotope records from North America, Liu et al. (2014b) indicated that there was a transition from a negative Pacific North American (PNA)-like pattern during the mid-Holocene to a positive PNA- like pattern during the late Holocene, which led to drier conditions in northwestern North America.

A similar conclusion was reached by Finkenbinder et al. (2016) based on lake sediment records from Newfoundland. They argued that this transition took place around 4.3ka B.P., leading to wetter conditions across the Newfoundland region. In contrast, Bond et al. (2001) argued that

North Atlantic SST anomalies around 4.2ka B.P. were related to a negative North Atlantic

Oscillation (NAO) pattern, linked to solar forcing. Deininger et al. (2017) also found that changes in the atmospheric circulation associated with northward and southward propagating westerlies (similar to the NAO but on a millennial instead of a decadal scale) could be a possible driver of coherency and cyclicity during the last 4.5ka BP, as seen in multiple speleothem $\delta^{18}O$ records that span most of the European continent. Thus, although there have been many suggested mechanisms, the ultimate drivers for climatic anomalies at 4.2ka B.P. remain unclear.

Wang (2009a) reviewed studies of Holocene cold events, and concluded that the most severe Holocene cold event, at ~8.2ka BP, was brought about by an outburst flood from pro-glacial

Lake Agassiz. This large volume of freshwater drained into the North Atlantic extremely rapidly, leading to a brief reorganization of the North Atlantic Meridional Overturning Circulation
(AMOC) and a southward displacement of the ITCZ, resulting in dry conditions over many regions
(Barber et al., 1999; Bianchi and McCave, 1999; Risebrobakken et al., 2003; McManus et al.,
2004; Clarke et al., 2004). Potential external forcing factors for the 4.2ka BP event include non-
linear responses to Milankovitch forcing, solar irradiance variations, and explosive volcanic
eruptions, all of which may have brought about variations in the ocean-atmosphere system (Booth
et al., 2005). Wang (2009a) concluded that solar irradiance minima were the main cause of cold
events in the mid- to late Holocene (including the 4.2ka BP event) and that internal oscillations
within the climate system could possibly have intensified these cold events under certain
circumstances (Wang, 2009b).

In summary, the 8.2ka BP event and corresponding southward shift in the ITCZ were
caused by glacial flooding of the North Atlantic and this can be reasonably simulated by coupled
GCMs with different boundary conditions and freshwater forcing (Alley and Agustsdottir, 2005;
LeGrande et al., 2006). By contrast, the forcing mechanisms that brought about the 4.2ka BP event
are currently uncertain. At 4.2ka B.P., the major global monsoon and ocean-atmosphere circulation
systems may have been deflected or weakened synchronously, causing major century-scale
precipitation disruptions, with severe megadroughts over many different regions (Weiss, 2017).
As GCM simulations of the 4.2ka BP event have not received much attention, in this study, the
spatial patterns and corresponding mechanisms relevant to the 4.2 ka BP event are examined and
compared to those associated with the 8.2ka BP event.

**2.    Data and methodology**
Simulations of the last 21ka (TRACE-21) were used in this study (He, 2011; He et al. 2013;
Wen et al., 2016). These transient simulations have been completed using Version 3 of the
Community Climate System Model (CCSM3), which is a coupled atmosphere-ocean general
circulation model developed by the National Center for Atmospheric Research (NCAR). The
atmosphere model in the CCSM3 is the Community Atmospheric Model 3 (CAM3) with a
horizontal resolution of ~3.75° (T31), and the ocean model is the Parallel Ocean Program (POP)
with a longitudinal resolution of 3.6° and variable latitudinal resolution.
The "full-forcing" TRACE-21 simulation includes changes in orbital parameters, greenhouse
gases, ice extent (based on the ICE 5G-VM2 configurations) and meltwater fluxes from the

Northern Hemisphere and Antarctic ice sheets. The orbital forcing is based on transient variations of orbital configuration (Berger, 1978). The concentrations of greenhouse gases were adopted from Joos and Spahni (2008). The ice sheet data were modified from the reconstruction of Peltier (2004) and the meltwater scheme was adopted from Liu et al. (2009).

Simulations in which only one of these factors was included have also been carried out and are available in the TRACE-21 archive (Otto-Bliesner et al., 2006; Wen et al., 2016). These simulations can reproduce the timing and magnitude of many aspects of climate evolution during the last 21 ka, such as changes in sea surface temperature (SST) (He et al., 2013). However, there are significant differences between the rate of temperature change in the model during the early Holocene and many paleoclimatic records (Liu et al., 2014a; Marcott et al., 2013; Marsicek et al., 2018). In this study, we do not address this enigma, but use the transient model data to compare intervals within the Holocene when abrupt changes in climate are known to have occurred in some regions (~8.2ka B.P. and ~4.2ka B.P.). These times were recently adopted by the International Commission on Stratigraphy as the chronological boundaries of the early, mid and late Holocene (Walker et al., 2012, 2018).

We examine mean annual surface temperature, annual precipitation and SSTs from the full-forcing experiment, and also AMOC strength, defined as the maximum Atlantic stream function between 20-50°N between 500m and 5000m depth (Ottera et al., 2010) from the full-forcing and orbital-forcing experiments.

**3. Results**

[revised manuscript text omitted]

Deleted: A possible explanation is that as summer insolation at high latitudes of the northern hemisphere declined over the Holocene, a threshold was passed which led to cooler SSTs in the North Atlantic and a consequent reduction in the Atlantic meridional overturning circulation (AMOC), with teleconnections into the southern hemisphere. In our experiment, we examined just the 5th millennium B.P., but it is possible that the changes seen in the latter half of the period were more persistent, and typical of the rest of the Holocene (the Neoglacial). Indeed, there is much evidence for cooler conditions and glacier expansion around the North Atlantic around this time (Solomina et al., 2015). When reviewing the glaciers in the Southern Hemishpere, the evidence found by Porter (2000) also support the concept about the onset of Neoglciation at mid-Holocene. The records of glacier fluctuations in Alaska also revealed that Neoglaciation began in some areas by 4.0 ka and major advances were underway by 3.0 ka, with two distinct early Neoglacial expansions centered on about 3.3-2.9 and 2.2-2.0 ka, respectively (Barclay et al., 2009). Thereafter, glaciers fluctuated but did not disappear again, indicating that a different climate state prevailedThis is distinctly different from the period prior to 5000 years B.P. when many mountain regions were ice-free.. Fluctuations around these cooler mean conditions may be related to internal centennial-scale ocean-atmosphere variability (cf. Wanner et al., 2011). This is distinctly different from the period prior to 5000 years B.P. when many mountain regions were ice-free. This is also confirmed by the AMOC strength anomalies after 4.8 ka BP from the all-forcing experiment and orbital-forc[ ... [1]
Moved (insertion) [2]
Moved up [2]: This is distinctly different from the period years B.P. when many mountain regions were ice-free.

It is clear that the earlier period was strongly influenced by freshwater forcing in the North Atlantic, which drastically reduced the Atlantic Meridional Overturning Circulation (AMOC). The similarity in anomaly patterns between the 8.2ka BP event and the late 5th millennium BP suggests that there was also disruption to the AMOC in the later period. However, as there was no comparable freshwater forcing in the 5th millennium B.P., we must therefore consider what other factors might have played a role in reducing AMOC strength. There were no major solar irradiance changes at that time, so we can rule that out as a forcing factor. However, there was a major eruption of the Icelandic volcano Hekla at ~4200 BP, and it is possible that such an event could have brought about regional cooling, leading to more extensive, thick sea-ice and attendant freshwater effects on the AMOC (cf. Moreno-Chamarro et al., 2017). This mechanism deserves further scrutiny.

In the "all forcing" TRACE-21 simulation, AMOC strength declined slightly during the late Holocene and underwent multi-century fluctuations (Fig. 7a), which were strongly correlated with SSTs in the region of the North Atlantic where cooling was so prominent from 4.5-4.0 ka B.P. (Fig. 8). Mean SSTs in this region over the last 4500 years of the model simulation stayed below the 4.8-4.5 ka B.P. average for ~69% of the time (Fig. 8), and AMOC strength was similarly below the 4.8-4.5 ka BP mean for 63% of the time (Fig. 7a). One of these fluctuations was associated with an AMOC minima around 4.2ka BP. In the TRACE-21 model simulation with only orbital forcing, AMOC strength reached its Holocene maximum around 4.8 ka BP, then slightly weakened (by ~10%) over the late Holocene, staying below the 4.8-4.5ka BP mean for 87% of the time, with minor multi-century variations superimposed on the long-term downward trend (Fig. 7b). This suggests that a combination of factors, led by long-term changes in insolation, drove a steady decline in SSTs across the North Atlantic and a reduction in the AMOC, with associated teleconnections across the globe (including drought in some regions). Minor fluctuations around this declining trend were the dominant pattern for most of the last 4500 years. This interpretation helps explain widespread paleoclimatic evidence for the onset of neoglaciation around 5000-4500 BP, followed by a series of neoglacial advances and retreats during recent millennia (Porter, 2000; Barclay et al., 2009; Solomina et al., 2015; Bradley and Bakke, 2018). Since the onset of neoglaciation early in the 5th millennium B.P., mountain glaciers fluctuated in extent but did not entirely disappear, indicating that a distinctly different climate state prevailed compared to the period prior to ~5 ka B.P., when many mountain regions were ice-free.

We therefore conclude from the model simulations that the "4.2ka B.P. event" was one of several late Holocene multi-century fluctuations that were embedded in a long-term, low frequency change in climate that occurred after ~4.8 ka BP. World-wide climatic anomalies during these fluctuations were driven by changes in the strength of the AMOC and related teleconnections. Whether such multi-century fluctuations were a response to internal centennial-scale ocean-atmosphere variability (cf Min and Liu, 2018), or external forcing (such as explosive volcanic eruptions and associated feedbacks) or a combination of such conditions, is not known. Further studies of the role of both external forcing and internal variability are needed to provide a better understanding of such mechanisms (cf. Ottera et al., 2010; Moreno-Chamarro et al., 2017; Gupta and Marshall, 2018).

**Acknowledgements**

This research was jointly supported by the National Key Research and Development Program of China (Grant No. 2016YFA0600401), the National Natural Science Foundation of China (Grant No. 41501210, Grant No. 41420104002, Grant No. 41671197, and Grant No. 41631175), the Jiangsu Province Natural Science Foundation (Grant No. BK20150977), Top-notch Academic Programs Project of Jiangsu Higher Education Institutions (Grant No. PPZY2015B115), and the Priority Academic Development Program of Jiangsu Higher Education Institutions (Grant No. 164320H116). Support was also received from U.S. NSF grant PLR-1417667 to the University of Massachusetts. TraCE-21ka was made possible by the DOE INCITE computing program, and supported by NCAR, the NSF P2C2 program, and the DOE Abrupt Change and EaSM programs.

[Figure]

**Figure 1**. (a) Northern hemisphere average surface temperature and (b) precipitation over the last 13ka years from the all-forcing experiment. Blue line is the 10-year running average and the black line is the 100-year running average. The black dash line shows the average of the time series.

[Figure]

[Figure]

[Figure]

**Figure 2**. The changes of (a) surface temperature (℃), (b) precipitation (mm/day), and (c) SST (℃) after 4.5ka BP (between 4500-4000ka BP and 4800-4500ka BP). The rectangles in (a) and (b) indicate the region with major dry-farming settlement abandonment around 4.2ka BP, according to Weiss (2016).

[Figure]

**Figure 3**. The first three patterns (a-c) and principal components (d-f) of rotated EOF modes on the SST over the period 4800-4000ka BP.

[Figure]

[Figure]

**Figure 4**. The changes of (a) surface temperature (°C), (b) precipitation (mm/day), and (c) SST (°C) after 8.8ka BP (between 8800-8000ka BP and 9200-8800ka BP)

[Figure]

**Figure 5**. The first three patterns (a-c) and principal components (d-f) of rotated EOF modes on the SST over the period 9200-8000ka BP.

[Figure]

[Figure]

**Figure 6**. The differences between changes of (a) surface temperature (°C), (b) precipitation (mm/day), and (c) SST (°C) of the 5$^{th}$ millennium BP and 9$^{th}$ millennium BP periods shown in Figures 2 and 4.

[Figure]

[Figure]

**Figure 7.** The 10-year running averaged (blue line) and 100-year running averaged (black line) time series of AMOC strength, plotted as anomalies from the mean for 4800-4500 years B.P. from (a) all-forcing experiment and (b) orbital-forcing experiment. AMOC strength was below the mean for 63% of the time in the all-forcing experiment and 87% of the time in the orbital-forcing experiment.

[Figure]

**Figure 8.** The 10-year running averaged (blue line) and 100-year running averaged (black line) SSTs in the area of the North Atlantic with significant negative SST differences between the 5th millennium BP and 9th millennium BP periods (40-60 °N, 7.5-60 °W) on Figure 2c, plotted as anomalies from the mean for 4800-4500 years B.P. ~69% of the time, temperatures in this region were below the mean.

A possible explanation is that as summer insolation at high latitudes of the northern hemisphere declined over the Holocene, a threshold was passed which led to cooler SSTs in the North Atlantic and a consequent reduction in the Atlantic meridional overturning circulation (AMOC), with teleconnections into the southern hemisphere. In our experiment, we examined just the 5th millennium B.P., but it is possible that the changes seen in the latter half of the period were more persistent, and typical of the rest of the Holocene (the Neoglacial). Indeed, there is much evidence for cooler conditions and glacier expansion around the North Atlantic around this time (Solomina et al., 2015). When reviewing the glaciers in the Southern Hemishpere, the evidence found by Porter (2000) also support the concept about the onset of Neoglciation at mid-Holocene. The records of glacier fluctuations in Alaska also revealed that Neoglaciation began in some areas by 4.0 ka and major advances were underway by 3.0 ka, with two distinct early Neoglacial expansions centered on about 3.3-2.9 and 2.2-2.0 ka, respectively (Barclay et al., 2009). Thereafter, glaciers fluctuated but did not disappear again, indicating that a different climate state prevailedThis is distinctly different from the period prior to 5000 years B.P. when many mountain regions were ice-free.. Fluctuations around these cooler mean conditions may be related to internal centennial-scale ocean-atmosphere variability (cf. Wanner et al., 2011). This is distinctly different from the period prior to 5000 years B.P. when many mountain regions were ice-free. This is also confirmed by the AMOC strength anomalies after 4.8 ka BP from the all-forcing experiment and orbital-forcing experiment, with ~63% and ~87% of the time below the mean in all-forcing experiment and orbital-forcing experiment (Fig. 8). Further analysis of the TRACE21 simulations are needed to fully explore this matter.

t seems clear that there was a fundamental shift in climate around this time. Furthermore, those changes have persisted, with minor fluctuations, through to the present. Interestingly, SSTs in the area of the North Atlantic where cooling was so prominent from 4500-4000 years B.P. do show multi-century-scale oscillations for the remainder late Holocene, with temperatures below the 4800-4500 years B.P. average for ~69% of the time (Fig. 7). Whether such changes are also linked to hydrological anomalies elsewhere, as with the period 4500-4000 years B.P., is not known, but it seems likely, given the large-scale coherent link between temperature and precipitation that is apparent in Fig. 1.

| Page 13: [3] Deleted | Raymond Bradley | 11/20/18 8:58 PM |
|---|---|---|

Nevertheless, we conclude from the model simulations that the "4.2ka B.P. event" was simply one of several late Holocene multi-century fluctuations that were embedded in a longer-term, lower frequency change in climate resulting from orbital forcing.

| Page 13: [4] Formatted | Raymond Bradley | 11/20/18 8:51 PM |
|---|---|---|

No widow/orphan control, Don't adjust space between Latin and Asian text, Don't adjust space between Asian text and numbers

| Page 13: [5] Formatted | Raymond Bradley | 11/22/18 6:53 PM |
|---|---|---|

Font:(Default) Times New Roman, 12 pt, Font color: Text 1

| Page 13: [6] Formatted | Raymond Bradley | 11/22/18 6:54 PM |
|---|---|---|

Justified, Indent: Left: 0", Hanging: 0.2", Line spacing: 1.5 lines

| Page 13: [7] Formatted | Raymond Bradley | 11/22/18 6:53 PM |
|---|---|---|

Font:(Default) Times New Roman, 12 pt, Font color: Text 1

| Page 13: [8] Formatted | Raymond Bradley | 11/22/18 6:55 PM |
|---|---|---|

Font:(Default) Times New Roman, 12 pt, Not Italic, Font color: Text 1

| Page 13: [9] Formatted | Raymond Bradley | 11/22/18 6:53 PM |
|---|---|---|

Font:(Default) Times New Roman, 12 pt, Font color: Text 1

| Page 13: [10] Formatted | Raymond Bradley | 11/22/18 6:53 PM |
|---|---|---|

Font:(Default) Times New Roman, 12 pt, Not Italic

| Page 16: [11] Formatted | Raymond Bradley | 11/22/18 6:53 PM |
|---|---|---|

Font:Italic, Font color: Text 1

| Page 16: [11] Formatted | Raymond Bradley | 11/22/18 6:53 PM |
|---|---|---|

Font:Italic, Font color: Text 1

| Page 16: [12] Formatted | Raymond Bradley | 11/22/18 6:53 PM |
|---|---|---|

Font:(Default) Times New Roman, 12 pt, Not Italic, Font color: Text 1

| Page 16: [12] Formatted | Raymond Bradley | 11/22/18 6:53 PM |
|---|---|---|

Font:(Default) Times New Roman, 12 pt, Not Italic, Font color: Text 1

| Page 16: [12] Formatted | Raymond Bradley | 11/22/18 6:53 PM |
|---|---|---|

Font:(Default) Times New Roman, 12 pt, Not Italic, Font color: Text 1

| Page 16: [13] Deleted | Raymond Bradley | 11/20/18 4:00 PM |
|---|---|---|

| Page 16: [13] Deleted | Raymond Bradley | 11/20/18 4:00 PM |
|---|---|---|

| Page 16: [14] Formatted | Raymond Bradley | 11/22/18 6:53 PM |
|---|---|---|

Font:Italic, Font color: Text 1

| Page 16: [14] Formatted | Raymond Bradley | 11/22/18 6:53 PM |
|---|---|---|

Font:Italic, Font color: Text 1

| Page 16: [15] Formatted | Raymond Bradley | 11/22/18 6:53 PM |
|---|---|---|

Font:Italic, Font color: Text 1

| Page 16: [15] Formatted | Raymond Bradley | 11/22/18 6:53 PM |
|---|---|---|

Font:Italic, Font color: Text 1

| Page 16: [16] Formatted | Raymond Bradley | 11/22/18 6:53 PM |
|---|---|---|

Font:Italic

| Page 16: [16] Formatted | Raymond Bradley | 11/22/18 6:53 PM |
|---|---|---|

Font:Italic

| Page 16: [17] Formatted | Raymond Bradley | 11/22/18 6:53 PM |
|---|---|---|

Font:Italic, Font color: Text 1

| Page 16: [17] Formatted | Raymond Bradley | 11/22/18 6:53 PM |
|---|---|---|

Font:Italic, Font color: Text 1

| Page 16: [18] Formatted | Raymond Bradley | 11/22/18 6:53 PM |
|---|---|---|

Font:Italic, Font color: Text 1

| Page 16: [18] Formatted | Raymond Bradley | 11/22/18 6:53 PM |
|---|---|---|

Font:Italic, Font color: Text 1

| Page 16: [18] Formatted | Raymond Bradley | 11/22/18 6:53 PM |
|---|---|---|

Font:Italic, Font color: Text 1

| Page 16: [19] Formatted | Raymond Bradley | 11/22/18 6:53 PM |
|---|---|---|

Font:(Default) Times New Roman, 12 pt

| Page 16: [19] Formatted | Raymond Bradley | 11/22/18 6:53 PM |
|---|---|---|

Font:(Default) Times New Roman, 12 pt

| Page 16: [19] Formatted | Raymond Bradley | 11/22/18 6:53 PM |
|---|---|---|

Font:(Default) Times New Roman, 12 pt

| Page 16: [19] Formatted | Raymond Bradley | 11/22/18 6:53 PM |
|---|---|---|

Font:(Default) Times New Roman, 12 pt

| Page 16: [19] Formatted | Raymond Bradley | 11/22/18 6:53 PM |
|---|---|---|

Font:(Default) Times New Roman, 12 pt

| Page 16: [20] Formatted | Raymond Bradley | 11/22/18 6:53 PM |
|---|---|---|

Font color: Text 1

| Page 16: [20] Formatted | Raymond Bradley | 11/22/18 6:53 PM |
|---|---|---|

Font color: Text 1

| Page 16: [21] Formatted | Raymond Bradley | 11/22/18 6:53 PM |
|---|---|---|

Font:Italic, Font color: Text 1

| Page 16: [21] Formatted | Raymond Bradley | 11/22/18 6:53 PM |

Font:Italic, Font color: Text 1

| Page 16: [22] Formatted | Raymond Bradley | 11/22/18 6:53 PM |

Font:Italic, Font color: Text 1

| Page 16: [22] Formatted | Raymond Bradley | 11/22/18 6:53 PM |

Font:Italic, Font color: Text 1

| Page 16: [23] Formatted | Raymond Bradley | 11/22/18 6:53 PM |

Font:Italic, Font color: Text 1

| Page 16: [23] Formatted | Raymond Bradley | 11/22/18 6:53 PM |

Font:Italic, Font color: Text 1

| Page 16: [24] Formatted | Raymond Bradley | 11/22/18 6:53 PM |

Font:Italic, Font color: Text 1

| Page 16: [24] Formatted | Raymond Bradley | 11/22/18 6:53 PM |

Font:Italic, Font color: Text 1

| Page 17: [25] Deleted | Raymond Bradley | 11/22/18 6:56 PM |

Thompson, L. G., Mosley-Thompson, E., Davis, M. E., Henderson, K. A., Brecher, H. H., Zagorodnov, V. S., Mashiotta, T. A., Lin, P.-N., Mikhalenko, V. N., Hardy, D. R., and Beer, J.: Kilimanjaro ice core records: evidence of Holocene climate change in tropical Africa, *Science*, 298, 589-593, 2002.

| Page 17: [26] Formatted | Raymond Bradley | 11/22/18 6:53 PM |

Font:(Default) Cambria Math, Font color: Text 1

| Page 17: [27] Formatted | Raymond Bradley | 11/22/18 6:54 PM |

Don't adjust space between Latin and Asian text, Don't adjust space between Asian text and numbers

| Page 17: [28] Deleted | Raymond Bradley | 11/20/18 3:08 PM |

Weiss, H.: Global megadrought, societal collapse and resilience at 4.2-3.9 ka BP across the Mediterranean and west Asia, *PAGES Magazine*, 24, 62-63, 2016.

| Page 17: [29] Formatted | Raymond Bradley | 11/22/18 6:53 PM |

Font color: R,G,B (34,34,34), Expanded by  0.15 pt

| Page 18: [30] Deleted | Raymond Bradley | 11/22/18 6:54 PM |

**Page 26: [31] Deleted**            **Raymond Bradley**            **11/20/18 9:19 PM**

[Figure]

[Figure]

**Figure 7.** The 10-year running averaged (blue line) and 100-year running averaged (black line) SSTs in the area of the North Atlantic with significant negative SST differences between the the 5[th] millennium BP and 9[th] millennium BP periods shown in dark blue (40-60 °N, 7.5-60 °W) on Figure 2c, plotted as anomalies from the mean for 4800-4500 years B.P.  ~69% of the time, temperatures in this region were below the mean.